# Diet Inflammatory Index among Regularly Physically Active Young Women and Men

**DOI:** 10.3390/nu16010062

**Published:** 2023-12-25

**Authors:** Anna Pietrzak, Anna Kęska, Dagmara Iwańska

**Affiliations:** 1Department of Human Biology, Józef Piłsudski University of Physical Education, 00-968 Warsaw, Poland; anna.pietrzak@awf.edu.pl; 2Department of Biomedical Sciences, Józef Piłsudski University of Physical Education, 00-968 Warsaw, Poland; dagmara.iwanska@awf.edu.pl

**Keywords:** physical activity, young adults, DII, nutrition

## Abstract

Recently, special attention has been paid to the relationship between diet and inflammation in the body. A factor that influences both diet and inflammation is physical activity. The aim of this study was to assess the inflammatory potential of the diets of young people engaging in regular physical activity. The participants were physical education students (*n* = 141 men and *n* = 151 women). The measurements included basic anthropometric parameters and a 4-day nutritional history from which the dietary inflammatory index (DII) was calculated. The average DII for female students was 2.09 ± 1.52, and that for male students was 0.21 ± 1.69. Consumption of all macro- and micronutrients was significantly higher among women and men with the lowest DII value (corresponding to an anti-inflammatory diet). The female and male students consuming anti-inflammatory diets were characterized by greater lean body mass (LBM), and, for the male students, a lower body fat content, compared to those whose diets were pro-inflammatory. Young and regularly physically active people are also exposed to the pro-inflammatory nature of their diets, whose long-term effects may lead to health problems.

## 1. Introduction

Adequate nutrition plays an important role in every period of a person’s life, influencing the processes of growth and maturation, achieving and maintaining a normal physique and physical fitness, and protecting against the consequences of ageing [1,2]. Nutrition is also attributed to the impact on the occurrence of many chronic non-communicable diseases (NCDs), which have long constituted the biggest health problem in many countries around the world [3,4]. Obesity, atherosclerosis and cardiovascular disease, many cancers, and respiratory and auto-immune disorders, which are among the main diseases in this regard, have a well-documented link to poor diet and the resulting abnormal body mass and chronic inflammation [5].

Diet can influence the risk of chronic disease through different mechanisms. The most common diet-related cause of the development of chronic inflammation in the body has been shown to be the consumption of excessive calories and the associated increase in body fat content. However, the composition of one’s diet, i.e., the types of food and nutrients consumed, is also important. It has been proven that diets based on foods rich in saturated fatty acids and sugars (e.g., the Western diet) contribute to a significant increase in body fat content as well as levels of indicators of inflammation (e.g., C-reactive protein (hsCRP)) in the blood. On the other hand, diets with a predominance of fruit, vegetables, vegetable oils, and fish (e.g., the Mediterranean diet) are conducive to reducing body weight and body fat as well as lowering levels of inflammatory biomarkers [6,7].

Research on the impact of diet on the occurrence of inflammation and the diseases that result from it has led to the development of the Dietary Inflammatory Index (DII), a new tool for analyzing diets for their inflammatory potential [8]. The DII, describing the impact of the entire diet, not individual nutrients, on inflammation, is consistent with the suggestions of many authors who believe that recommendations focused on the consumption of several food groups are, in practice, more important than recommendations focused on specific foods or nutrients [9,10]. Based on numerous publications released between 1950 and 2007, inflammation score values for products or nutrients have been developed. Those that have obtained a positive index are characterized by pro-inflammatory substances (e.g., saturated fatty acids), while negative values indicate anti-inflammatory substances (e.g., vitamins and minerals) [8,11]. The unification of the DII allowed the assessment of the quality of food consumed, as well as the assessment of its impact on health. It has been proven that DII values correlate with indicators of inflammation (e.g., CRP, fibrinogen, and Interleukin-6 (IL-6)) [12] and that the DII is a better index for evaluating the impact of diet on all-cause mortality than other diet scores, e.g., HEI-2015 [13]. 

The previous studies on the DII available in the literature mainly concern people in middle or late adulthood, often in the initial or advanced stages of disease (e.g., atherosclerosis, diabetes, and cancers) [14,15,16]. There are still not many studies dedicated to dietary inflammatory potential, determined using the DII, among young, healthy individuals, especially those engaging in regular physical activity. Physically active people, compared to those with sedentary lifestyles, have higher requirements for energy and many nutrients, i.e., carbohydrates and proteins, as well as many vitamins and minerals [17,18]. However, it should be emphasized that high physical performance is closely linked to a specific body structure, i.e., high muscle mass and low body fat [19,20]. Therefore, the diets consumed by regularly exercising people, despite the increased calorie content, should allow them to maintain proper body mass and composition. In addition, the diets of physically active people should offset the negative effects of exercise, i.e., enable the repair of muscle fibers damaged during exercise and reduce exercise-induced oxidative stress, which contribute to post-exercise inflammation. Thus, assessing the impact of diet as an essential part of exercise preparation and post-exercise recovery on body composition and inflammation among active subjects is essential in the context of the health benefits of an active lifestyle, including the prevention of chronic non-communicable diseases. The aim of this study was to assess the inflammatory potential of the diets of young adults engaging in regular physical activity. This study was planned to be conducted on healthy individuals of both sexes whose physical activity was of a moderate but regular nature, i.e., the degree considered to yield the greatest health benefits.

## 2. Materials and Methods

### 2.1. Subjects

Study participants were first- and second-year physical education students enrolled at the University of Physical Education in Warsaw. The participants, due to their field of study, engaged in regular physical activity like running, swimming, and team games (about 9 h/week). A total of 292 young adults were deemed eligible to take part in the study, including 151 women and 141 men. They were healthy people who did not take any medication on a regular basis were and non-smokers. Additional inclusion criteria were age < 25 years, body fat content < 32% in women and <25% in men [21], non-engagement in competitive sport, and provision of consent to participate in the study.

Each participant was informed about the purpose and procedures of the research and provided their written consent to it. The research protocol was approved by the Ethics Committee of the Józef Piłsudski University of Physical Education in Warsaw (No. SKE 01-18/2017).

### 2.2. Anthropometric Measurements

Body weight and body height were measured using an electronic scale with a stadiometer (model Seca 799, manufactured by Seca Gmbh & Co. Kg, Hamburg, Germany). Body height measurements were taken to the nearest 0.5 cm, and body weight measurements were taken to the nearest 0.1 kg. Body mass index (BMI) was calculated using the following formula: weight (kg)/height (m)^2^. Body fat content was determined using the Durnin and Womersley method [22]. For this purpose, the thicknesses of four skinfolds (biceps, triceps, suprailiac, and subscapular) were measured using a Harpenden Skinfold Caliper (British Indicators, Burgess Hill, UK). Body fat mass and lean body mass (LBM) were also calculated.

Anthropometric measurements were taken twice, and the results were then averaged. All measurements were taken by a trained researcher on the morning of the same day, with the subjects wearing sports outfits without shoes.

### 2.3. Dietary Assessment

Total energy and macronutrient and micronutrient intake were assessed using dietary notes provided by each participant; the participants were instructed prior to the study on how to complete this task. Dietary information was collected during a typical week for the subjects in terms of both nutrition and physical activity. Dietary information included all meals, drinks and supplements consumed over 4 days (2 weekdays and 2 weekend days). Students’ nutritional notes were then validated by a trained interviewer who established the exact size of meals using a set of pictures of meals and food. Energy intake and diet composition were analyzed using the computer program ‘Dieta 5.0’ purchased from the National Food and Nutrition Institute (Warsaw, Poland).

### 2.4. Dietary Inflammatory Index

The method reported by Shivappa et al. was used to determine the DII values of the study participants’ diets [23]. Shivappa et al. identified 45 food components that affect the concentration of inflammatory or anti-inflammatory biomarkers. In this study, the DII score was calculated on the basis of 30 food components (instead of 45) due to the lack of some nutrients (such as ginger, turmeric, garlic, oregano, hot pepper, rosemary, eugenol, saffron, flavan-3-ol, flavones, flavonols, flavonones, and anthocyanidins) in the nutrient database of the computer program ‘Dieta 5.0’. To calculate DII for the participants of this study, the dietary data were linked to the standard global mean as a z-score. All of the food-parameter-specific DII scores were then summed to obtain the overall DII score for every participant in the study.

### 2.5. Statistical Analysis

Groups were divided by the DII values into quartiles. All variables were checked for normality using the Shapiro–Wilk test. The statistical significance of the differences between three groups was assessed using Kruskal–Wallis ANOVA. Data are presented as means ± SD. *p* < 0.05 was considered significant. Correlations between anthropometric parameters and the DII were examined by assessing Spearman’s simple correlation coefficients. In addition, in order to illustrate the changes in physique under the influence of DII, the results of the selected variables were approximated using the simple regression equation y = a + bt. The b coefficient reflected the rate of change of a given variable as a function of the DII. All calculations were carried out using Statistica v.13. (StatSoft, Palo Alto, CA, USA).

## 3. Results

The overall characteristics of the study population are shown in Table 1. As expected, female students had significantly lower height and weight values than male students and a significantly higher body fat content.

The amount of energy and the quantities of selected nutrients consumed by the female and male students are shown in Table 2 and Table 3. Only DII components found in the participants’ diets are included in these tables. The mean DII score for women was 2.09 ± 1.52, and that for men was 0.21 ± 1.69. Consumption of all macro- and micronutrients was significantly higher among women and men with the lowest DII values (Q1—the most anti-inflammatory diet) (Table 2 and Table 3). Female students whose diets were more pro-inflammatory did not differ from their peers with an anti-inflammatory diet, except in terms of cholesterol intake (Table 2), while male students following diets with different DIIs did not differ significantly in saturated fatty acid consumption (Table 3).

When comparing the physiques and body compositions of the male and female participants divided into quartile groups according to the DII, it was observed that those with lower DII (corresponding to an anti-inflammatory diet) values were significantly taller and heavier compared to the other study participants. Moreover, female students in group Q1 (with the lowest DII) differed significantly from female students in group Q4 (with the highest DII) only in lean body mass, whereas body fat content was similar (Table 4). Among male students, those whose diets were the most anti-inflammatory (group Q1) had significantly higher LBM and significantly lower body fat compared to those consuming proinflammatory diets (group Q4), although the BMI values of the men included in the distinguished quartile groups did not differ (Table 5).

For both sexes, there were significant associations between an index describing the inflammatory potential of the diet and body weight and LBM (positive correlations) as well as body fat percentage (negative correlation) (Table 6). When comparing the effects of the DII on the main body components of the study participants, it was found that the DII first leads to changes in LBM and only then in body fat. The results of the regression analysis shown in Figure 1 and Figure 2 confirmed that differences in the DII are first responsible for changes in LBM and only then in body fat content.

## 4. Discussion

In this study, the range of DII values for the female students was from −2.6 to 4.36, while in the male students, it was from −3.39 to 4.23. According to The US News and World Report (USNWR), which publishes an annual ranking of the most popular diets and an assessment of their nutritional value, the diets with the highest anti-inflammatory potential, i.e., the macrobiotic diet, the Biggest Loser diet, and the Ornish diet, had a DII index below −4 (−4.82, −4.85, and −4.06, respectively). In contrast, the diets with the highest pro-inflammatory potential, i.e., the Dukan, South Beach, and keto diets, had DII values of +3.42, +1.39, and +0.84, respectively [24]. Relating the results of the present study to these data, it can be concluded that the diets of some physical education students had a strong pro-inflammatory potential, comparable to that attributed to diets with documented adverse health effects.

Young, non-obese subjects of both sexes (78 men and 26 women) with moderate physical activity levels were also studied by Jagielski et al. [25]. The mean DII value for the participants in this study was −0.82, with a range between −2.78 and 2.93 (no separate values are given for men and women). Upon comparing this finding with our own results (wherein the mean DII value for all 292 participants was 1.17 ± 1.85, with a range of −3.39–4.36), it can be seen that there were more pro-inflammatory diet adherents among female and male physical education students than in the population studied by Jagielski et al. It is likely that this difference can partly be explained by the fact that the participants in the present research were younger than those studied by Jagielski et al. (with a mean age of 34.66 ± 5.76). It is worth noting that Jagielski et al., in their study, showed that young people who followed a balanced diet may have an 86% lower risk of contracting COVID-19 compared to those adhering to a higher-DII diet [25]. As the result concerned a population similar to ours, it can be assumed that adherence to a pro-inflammatory diet for some physical education students adversely affects their immunity.

The high DII values observed for some participants in the present study confirm reports by other authors concerning a high frequency of poor eating behaviors among young adults [26,27,28]. At this point, it should be emphasized that the participants in this study were first- and second-year students. Starting university is often associated with a change of residence or social environment and thus with significant lifestyle changes, e.g., regarding diet, physical activity, and use of stimulants [29]. It has been shown that these changes often lead to unhealthy behaviors, e.g., high consumption of sweet snacks and high-fat foods, low consumption of fruit and vegetables, and high alcohol intake. This results in an increase in body weight and body fat content and unfavorable remodeling of the metabolic profiles of young adults [30,31]. This seems to be particularly problematic for physically active people, in whom inflammation can also result from exercise. It has been proven that depending on intensity and duration and thus the magnitude of muscle damage and exercise-induced oxidative stress, physical effort can induce different increases in inflammatory markers [32]. Thus, if an exerciser’s diet is pro-inflammatory, it not only hinders the amelioration of post-exercise inflammation but also reduces the health benefits associated with an active lifestyle.

There is a large body of literature showing that regular, moderate physical activity can lower resting levels of inflammatory markers, which may be useful, especially for people suffering from obesity and other diseases associated with chronic inflammation [33,34]. However, it is stressed that the health benefits of regular exercise are highly dependent on adherence to a proper diet, which should, above all, provide sufficient energy [35,36]. The present study showed that both female and male students whose diets were the least inflammatory (Q1 groups) had significantly higher calorie intakes than the others. In addition, it was found that the daily food intake in all quartiles of women and in quartiles Q2–Q4 of men was lower than their energy requirements [37]. Insufficient caloric intake is a common phenomenon among athletes, and a persistent negative energy balance may induce many health problems in active people. Specifically, it has been shown that low energy availability is associated with hormonal disturbances, disruption of bone metabolism, decreased immune function, and changes in the skeletal and other systems, which are collectively described in the literature as RED-S (Relative Energy Deficiency in Sport) syndrome [38]. Thus, the results of our study confirm that negative energy balance is common not only among professional athletes but also among young physically active people. Furthermore, it has been shown that a diet containing too few calories in relation to dietary requirements can also have a pro-inflammatory effect. On the other hand, it should be recalled that energy intake is one of components in DII scores that adopts pro-inflammatory values for excessively high energy consumption [8]. This means that neither an overly high nor overly low energy intake is unsuitable for the human body.

The female and male students whose diets were anti-inflammatory had significantly higher intakes of all analyzed macro- and micronutrients. The only component whose intake was similar among female students consuming diets with different DII values was cholesterol. Eggs, meat, and dairy products are foods that contain dietary cholesterol. These products are also a source of high-quality protein, a higher intake of which is recommended for physically active people [39]. The lack of differences in cholesterol intake between the groups of female students, who, however, differed in terms of protein intake, indicates that their diets contained higher proportions of other products providing this nutrient. Such products include fish, legumes, and nuts, which are considered to have a beneficial effect on DII values. Meat and dairy, but not eggs, in addition to cholesterol, are also sources of SFA [40]. Male students did not differ in terms of SFA consumption, but it was observed that those from the Q1 group consumed significantly more cholesterol compared to the other groups. Just like in the case of female students, these differences resulted from similar or different intakes of products that are the sources of these ingredients.

As a consequence of differences in the amount of food consumed, differences in physique and body composition were noted between the distinguished quartile groups of the female and male students. It was observed that female and male students consuming the highest-calorie but least-inflammatory diets (Q1 groups) were characterized by the highest body weight. It is interesting to note that for both sexes, participants in the Q1 groups were primarily characterized by significantly greater LBM compared to those in the Q4 group (*p* < 0.001). In addition, male students consuming mostly pro-inflammatory diets (Q4 group) had significantly more body fat (*p* < 0.05), while among the female students, body fat did not differ between the Q1 and Q4 groups. The differences in LBM and body fat in physically active young people are in agreement with other authors’ data relating that the beneficial effects of systematic physical activity on body composition may be limited by an inappropriate diet [41]. It may seem surprising that the study participants with a lower calorie intake had more body fat. However, given that they were all lean individuals with increased energy expenditure, it can be ruled out that their higher body fat content was due to the activation of energy-saving mechanisms. Other authors also drew attention to the occurrence of such a phenomenon among physically active people [42].

Another important observation made in this study is that the DII correlates significantly with both LBM and body fat in both female and male students and that in both sexes, this relationship was stronger for LBM. This is of particular relevance to physically active people, as it means that a pro-inflammatory diet makes it more difficult to increase LBM, of which muscle is a major component, thus making it a significant factor in reducing their exercise capacity. In addition, there have been studies indicating that reduced lean mass/muscle mass in healthy adult men (18–34 years) is associated with reduced insulin sensitivity [43,44].

The present study has some limitations to consider. First, no method of invasive examination of inflammation (e.g., markers of inflammation in the blood) was used in the methodology of this study. We are not completely certain that the inflammation resulting from the DII questionnaire actually occurs in people with the highest DII score, but there are a lot of publications showing connections with DII scores and inflammatory markers. Second, nutritional notes were made according to 4 days of consumption; therefore, the actual dietary intake of the subjects can be averaged. Third, the size of the study population was modest. The advantages of this study, on the other hand, include the homogeneity of the study population and the fact that the information on the diets of the study subjects was collected using dietary notes.

## 5. Conclusions

This study provides insight into the pro-inflammatory potential of the diets of young, healthy, and physically active individuals, a population that has rarely been analyzed to date. Its results show that a group of physically active people followed a diet that reduces the benefits of regular exercise. This may be due to insufficient calorie intake, which, as shown in this study, increases the pro-inflammatory nature of a diet for both females and males. An important finding of this study is that a pro-inflammatory diet (confirmed by DII values), when consumed by young, lean individuals, primarily reduces the proportion of lean body mass in body composition. The results show the need for the more effective education of young people, especially those leading active lifestyles, regarding proper nutrition.

## Figures and Tables

**Figure 1 nutrients-16-00062-f001:**
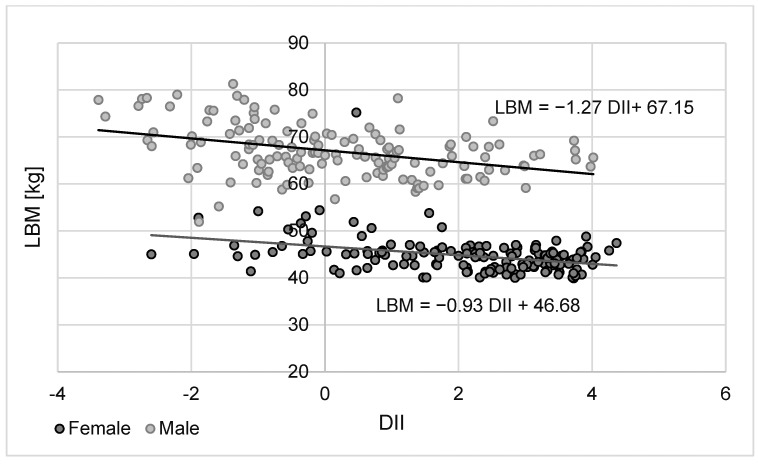
Changes in LBM values as a function of DII approximated using a simple regression equation for female and male students.

**Figure 2 nutrients-16-00062-f002:**
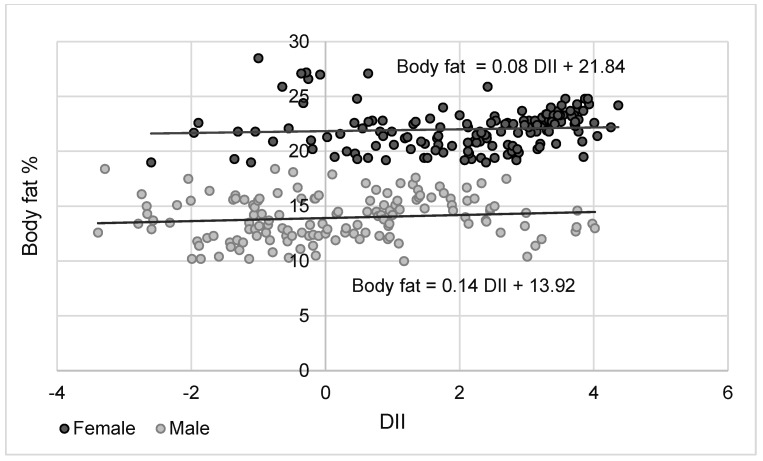
Changes in body fat values as a function of DII approximated using a simple regression equation for female and male students.

**Table 1 nutrients-16-00062-t001:** Anthropometric characteristics of study participants (mean ± SD).

	Female Students (*n* = 151)	Male Students (*n* = 141)
DII (min; max)	(−2.60; 4.36)	(−3.39; 4.23)
Age (years)	19.7 ± 1.12	22.1 ± 1.60
Height (cm)	165.5 ± 4.03 ^a^	181.1 ± 6.08
Body mass (kg)	57.2 ± 4.71 ^a^	77.8 ± 6.70
BMI *	20.9 ± 1.01 ^a^	23.7 ± 1.68
Body fat%	22.0 ± 1.90 ^a^	13.9 ± 2.05
Body fat (kg)	12.7 ± 2.01 ^a^	10.9 ± 2.05
LBM ^^^ (kg)	44.8 ± 3.90 ^a^	66.9 ± 5.45

* Body Mass Index; ^^^ Lean Body Mass; ^a^ *p* < 0.001 versus active males.

**Table 2 nutrients-16-00062-t002:** Daily energy, macronutrient, and micronutrient intake among female students (mean ± SD).

	Q1 (*n* = 38) Most Anti-Inflammatory Diet	Q2–Q3 (*n* = 75)	Q4 (*n* = 38) Most Pro-Inflammatory Diet
DII (min; max)	(−2.60; 1.02)	(1.18; 3.29)	(3.30; 4.36)
Energy (kcal)	2221.0 ± 718.94 ^a,b^	1669.3 ± 382.08	1463.0 ± 417.65
Protein (g)	73.8 ± 21.91 ^a,b^	58.1 ± 14.19	51.8 ± 14.27
Fat (g)	88.9 ± 39.90 ^c^	66,0 ± 20.95	58.7 ± 21.94
Carbohydrates (g)	305.5 ± 87.77 ^a,b^	220.6 ± 54.04	190.7 ± 62.71
Magnesium (mg)	340.3 ± 218.61 ^a,b^	246.3 ± 278.02 ^d^	180.0 ± 41.78
Iron (mg)	12.2 ± 3.24 ^a,b^	8.8 ± 1.85 ^d^	7.2 ± 2.03
Zinc (mg)	10.3 ± 3.08 ^a,b^	7.6 ± 1.72 ^d^	6.6 ± 1.61
Vitamin A (µg)	1315.8 ± 650.80 ^a,b^	593.4 ± 251.42 ^d^	447.4 ± 176.11
Beta carotene (µg)	5421.9 ± 3988.44 ^a,b^	1686.1 ± 1333.94 ^d^	833.0 ± 447.35
Vitamin E (mg)	13.6 ± 6.23 ^e,b^	9.8 ± 3.33 ^b^	6.7 ± 3.28
Thiamine (mg)	1.16 ± 0.37 ^a,b^	0.83 ± 0.39 ^b^	0.58 ± 0.16
Riboflavin (mg)	1.55 ± 0.49 ^a,b^	1.20 ± 0.39 ^b^	1.05 ± 0.32
Niacin (mg)	13.4 ± 4.25 ^b,c^	10.5 ± 2.97 ^b^	8.1 ± 2.70
Vitamin B6 (mg)	1.69 ± 0.45 ^a,b^	1.24 ± 0.31 ^b^	0.87 ± 0.23
Vitamin C (mg)	70.6 ± 39.27 ^b,e^	46.8 ± 28.20 ^b^	28.1 ± 14.97
Saturated fatty acids (g)	31.3 ± 14.27 ^d,e^	23.7 ± 8.26	23.0 ± 9.65
Monounsaturated fatty acids (g)	35.1 ± 17.06 ^b^	26.3 ± 8.62	23.1 ± 8.95
Polyunsaturated fatty acids (g)	15.7 ± 8.91 ^b^	11.6 ± 4.54 ^b^	8.1 ± 4.19
Cholesterol (mg)	312.0 ± 169.06	250.2 ± 95.95.83	234.7 ± 104.11
Dietary fiber (g)	22.2 ± 5.25 ^a,b^	14.2 ± 3.31 ^b^	10.2 ± 2.54

^a^ significant differences vs. Q2–Q3, *p* < 0.001; ^b^ significant differences vs. Q4, *p* < 0.001; ^c^ significant differences vs. Q2–Q3, *p* < 0.01; ^d^ significant differences vs. Q4, *p* < 0.05; ^e^ significant differences vs. Q2–Q3, *p* < 0.05.

**Table 3 nutrients-16-00062-t003:** Daily energy, macronutrient, and micronutrient intake in male students (mean ± SD).

	Q1 (*n* = 36) Most Anti-Inflammatory Diet	Q2–Q3 (*n* = 69)	Q4 (*n* = 36) Most Pro-Inflammatory Diet
DII (min; max)	(−3.39; −1.07)	(−1.02; 1.30)	(1.36; 4.23)
Energy (kcal)	3700.8 ± 781.58 ^A,B^	2902.9 ± 559.43 ^C^	2398.6 ± 429.73
Protein (g)	133.7 ± 25.35 ^A,B^	103.5 ± 21.33 ^B^	79.9 ± 14.91
Fat (g)	151.7 ± 39.56 ^A,B^	117.8 ± 34.21 ^C^	100.8 ± 21.94
Carbohydrates (g)	486.8 ± 106.61 ^A,B^	377.8 ± 72.78 ^B^	306.1 ± 71.41
Magnesium (mg)	463.5 ± 80.53 ^A,B^	361.4 ± 67.30 ^B^	255.7 ± 57.62
Iron (mg)	19.4 ± 3.01 ^A,B^	15.1 ± 3.86 ^B^	10.9 ± 2.92
Zinc(mg)	17.8 ± 3.11 ^A,B^	13.5 ± 2.52 ^B^	10.7 ± 2.07
Vitamin A (µg)	1851.8 ± 1021.96 ^A,B^	1027.8 ± 496.28 ^C^	751.5 ± 250.65
Beta carotene (µg)	6064.9 ± 5775.52 ^A,B^	2614.2 ± 2419.70 ^C^	1467.9 ± 872.58
Vitamin E (mg)	22.0 ± 7.64 ^A,B^	16.3 ± 5.78 ^B^	11.3 ± 4.21
Thiamine (mg)	1.96 ± 0.40 ^A,B^	1.55 ± 0.39 ^B^	1.08 ± 0.24
Riboflavin (mg)	2.47 ± 0.63 ^A,B^	1.83 ± 0.48 ^C^	1.47 ± 0.37
Niacin (mg)	25.8 ± 6.68 ^D^	21.4 ± 7.78 ^B^	14.0 ± 4.08
Vitamin B6 (mg)	2.95 ± 0.57 ^A,B^	2.22 ± 0.50 ^B^	1.43 ± 0.34
Vitamin C (mg)	90.9 ± 39.38 ^A,B^	53.2 ± 28.06 ^B^	34.6 ± 20.75
Saturated fatty acids (g)	51.7 ± 16.90	39.9 ± 14.65	38.6 ± 8.53
Monounsaturated fatty acids (g)	60.9 ± 16.79 ^A,B^	48.1 ± 15.44 ^C^	40.0 ± 9.48
Polyunsaturated fatty acids (g)	27.1 ± 10.08 ^B^	20.8 ± 7.42 ^B^	14.3 ± 5.54
Cholesterol (g)	620.9 ± 266.70 ^B^	494.7 ± 247.62	399.1 ± 137.61
Dietary fiber (g)	32.9 ± 8.59 ^A,B^	23.3 ± 5.02 ^B^	16.2 ± 4.60

^A^ significant differences vs. Q2–Q3, *p* < 0.001; ^B^ significant differences vs. Q4, *p* < 0.001; ^C^ significant differences vs. Q4, *p* < 0.01; ^D^ significant differences vs. Q2–Q3, *p* < 0.01.

**Table 4 nutrients-16-00062-t004:** Comparison of the physiques and body compositions of female students divided into quartile groups according to the DII (mean ± SD).

	Q1 (*n* = 38) Most Anti-Inflammatory Diet	Q2–Q3 (*n* = 75)	Q4 (*n* = 38) Most Pro-Inflammatory Diet
DII (min; max)	(−2.60; 1.02)	(1.18; 3.29)	(3.30; 4.36)
Height (cm)	167.8 ± 5.58 ^a,d^	164.7 ± 2.92	165.0 ± 3.25
Body mass (kg)	60.3 ± 6.42 ^a,d^	56.0 ± 3.54	56.6 ± 3.30
BMI *	21.3 ± 1.30 ^c,f^	20.6 ± 0.90	20.8 ± 0.60
Body fat%	22.3 ± 2.69 ^e^	21.4 ± 1.40 ^g^	23.0 ± 1.17
Body fat (kg)	13.6 ± 3.05 ^a^	12.0 ± 1.33 ^h^	13.0 ± 1.27
LBM (kg) ^^^	47.4 ± 5.87 ^a,b^	44.0 ± 2.49	43.6 ± 6.77

* Body Mass Index; ^ Lean Body Mass; ^a^ significant differences vs. Q2–Q3, *p* < 0.001; ^b^ significant differences vs. Q4, *p* < 0.001; ^c^ significant differences vs. Q2–Q3, *p* < 0.01; ^d^ significant differences vs. Q4, *p* < 0.01; ^e^ significant differences vs. Q2–Q3, *p* < 0.05; ^f^ significant differences vs. Q4, *p* < 0.05; ^g^ significant differences vs. Q4, *p* < 0.001; ^h^ significant differences vs. Q4, *p* < 0.01.

**Table 5 nutrients-16-00062-t005:** Comparison of the physiques and body compositions of male students divided into quartile groups according to the DII (mean ± SD).

	Q1 (*n* = 36) Most Anti-Inflammatory Diet	Q2–Q3 (*n* = 69)	Q4 (*n* = 36) Most Pro-Inflammatory Diet
DII (min; max)	(−3.39; −1.07)	(−1.02; 1.30)	(1.36; 4.23)
Height (cm)	184.2 ± 6.32 ^C,D^	180.6 ± 4.84	179.1 ± 6.56
Body mass (kg)	82.3 ± 8.79 ^A,B^	76.8 ± 5.26	75.2 ± 4.21
BMI *	24.2 ± 2.03	23.6 ± 1.50	23.5 ± 1.57
Body fat%	13.5 ± 2.17 ^E,F^	13.8 ± 1.94 ^I^	14.7 ± 1.76
Body fat (kg)	11.2 ± 2.61	10.6 ± 1.95	11.1 ± 1.49
LBM (kg) ^^^	71.1 ± 6.79 ^A,B^	66.2 ± 4.08 ^H^	64.1 ± 3.67

* Body Mass Index; ^ Lean Body Mass; ^A^ significant differences vs. Q2–Q3, *p* < 0.001; ^B^ significant differences vs. Q4, *p* < 0.001; ^C^ significant differences vs. Q2–Q3, *p* < 0.01; ^D^ significant differences vs. Q4, *p* < 0.01; ^E^ significant differences vs. Q2–Q3, *p* < 0.05; ^F^ significant differences vs. Q4, *p* < 0.05; ^H^ significant differences vs. Q4, *p* < 0.01; ^I^ significant differences vs. Q4, *p* < 0.05.

**Table 6 nutrients-16-00062-t006:** Spearman’s simple correlation coefficients for anthropometric parameters and DII.

	Female Students	Male Students
	DII	
Body mass (kg)	−0.181	−0.338
	*p* < 0.05	*p* < 0.001
BMI	−0.110	−0.161
	*p* = 0.178	*p* = 0.057
Body fat %	0.286	0.178
	*p* < 0.001	*p* < 0.05
Body fat (kg)	0.081	−0.010
	*p* = 0.324	*p* = 0.902
LBM (kg)	−0.316	−0.395
	*p* < 0.001	*p* < 0.001

## Data Availability

The data presented in this study are available on request from the corresponding author. The data are not publicly available due to confidentiality reasons.

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
