# Peer review of "Diet Inflammatory Index among Regularly Physically Active Young Women and Men"

_nutrients, 2023, doi:10.3390/nu16010062_

Round 1
Reviewer 1 Report
Comments and Suggestions for Authors
Technical aspect:
• English language correction is required;
• Careful checking of text for letters’ size and type, but also for misplaced parts of sentences, is required;
• Please check and correct references list for referencing style, but also references on the list not cited in the main text;
Merit aspects:
Key words:
I believe that key word” inflammation “ is misleading in the context of this paper: please remove this key word;
Introduction:
• 2nd paragraph:” Consuming excessive dietary calories…” this sentence in the present form has no sense: rephrase the sentence;
• Once authors state Dietary Inflammatory Index, in the same paragraph Inflammatory Diet Index – unify the name and use it consequently;
• DII correlates with which indicators of inflammation: please give additional information;
Material and Methods:
• Subjects: a number of 440 students is not important to any part of this study: remove this information from MM section;
• Standard medical equipment – please give names and manufacturer data;
• This is my main merit objection: Body fat and lean body mass (LBM) was in this study assessed with very old and not reliable method and referenced with paper from 1974: I studied this paper carefully: authors could calculate subcutaneous fat with skin fold measurements, but not LBM because this method shows no way to estimate visceral fat (very important from immune point of view); Because of this fact, some of the further presented results concerning body fat % and LBM and big parts of discussion are not reliable too. In the previous paper (Int J Eniviron Res Public Health 2022: 19: 6884) authors used Tanita scale to acquire fat content and LBM with bioimpedance method, which is widely used and accepted for body fat content and LBM estimation also in scientific literature – why not in this study- please give reliable measurements of body fat and LBM;
Results:
• Authors can not compare effects of DII (DII not DII index as the name includes word ”index“) on body components as there is not enough data: only a single measurement of body composition was performed (not reliable measurement) and dietary data from only 1 week (even less: 4 days) were collected. Please remove this part of the paragraph.
• Authors haven’t present any results or parameters suggesting that diet influenced the inflammatory status of the participants (for instance: plasma levels of inflammatory markers before and after diet usage – authors mentioned about this fact in limitations of the study paragraph); please remove this statement;
Discussion:
• ‘’…diet of some students favored an increase in inflammatory markers” this sentence has no reflection in the results – remove or rephrase the sentence;
• It would be better not to make any assumptions concerning immunity, as, again, authors haven’t measured any parameters concerning immune system functioning.
• There are some results which are presented in results section but not discussed in discussion section: lack of differences in cholesterol consumption in females and saturated fatty acids in males – supplement the discussion;
Minor language correction is required as there are some spelling mistakes.
Author Response
Answer to Reviewer 1
We would like to thank the reviewer for his time and effort in reviewing our work and for all the feedback on the manuscript we submitted. We are grateful for all comments, which we have carefully considered. Please find below our responses/clarifications.
Technical aspect:
- English language correction is required;
• Careful checking of text for letters’ size and type, but also for misplaced parts of sentences, is required;
• Please check and correct references list for referencing style, but also references on the list not cited in the main text;
Authors’ reply: The font was standardised and the manuscript was checked by a professional as suggested by the reviewer. The list of references has been standardised and checked for citation in the main text.
Merit aspects:
Key words:
I believe that key word” inflammation “ is misleading in the context of this paper: please remove this key word;
Authors’ reply: As suggested by the reviewer, the key words have been changed.
Introduction:
• 2nd paragraph: ”Consuming excessive dietary calories…” this sentence in the present form has no sense: rephrase the sentence;
Authors’ reply: This sentence has been changed.
- Once authors state Dietary Inflammatory Index, in the same paragraph Inflammatory Diet Index – unify the name and use it consequently;
Authors’ reply: Thank you for this comment. The name of the indicator has been corrected throughout the manuscript.
- DII correlates with which indicators of inflammation: please give additional information;
Authors’ reply: This information has been completed.
Material and Methods:
• Subjects: a number of 440 students is not important to any part of this study: remove this information from MM section;
Authors’ reply: This information has been removed.
- Standard medical equipment – please give names and manufacturer data;
Authors’ reply: This information has been completed.
- This is my main merit objection: Body fat and lean body mass (LBM) was in this study assessed with very old and not reliable method and referenced with paper from 1974: I studied this paper carefully: authors could calculate subcutaneous fat with skin fold measurements, but not LBM because this method shows no way to estimate visceral fat (very important from immune point of view); Because of this fact, some of the further presented results concerning body fat % and LBM and big parts of discussion are not reliable too. In the previous paper (Int J Eniviron Res Public Health 2022: 19: 6884) authors used Tanita scale to acquire fat content and LBM with bioimpedance method, which is widely used and accepted for body fat content and LBM estimation also in scientific literature – why not in this study- please give reliable measurements of body fat and LBM;
Authors’ reply: We agree with the reviewer that the assessment of body composition based on the measurement of skinfold thickness is a method that has been used for a long time. However, it is still a method practised by many researchers, not only by us (Pérez-Chirinos Buxadé C, Solà-Perez T, Castizo-Olier J, Carrasco-Marginet M, Roy A, Marfell-Jones M, Irurtia A. Assessing subcutaneous adipose tissue by simple and portable field instruments: Skinfolds versus A-mode ultrasound measurements. PLoS One. 2018, 13(11):e0205226; Szeszulski J, Lorenzo E, Arriola A, Lee RE. Community-Based Measurement of Body Composition in Hispanic Women: Concurrent Validity of Dual- and Single-Frequency Bioelectrical Impedance. J Strength Cond Res. 2022, 36(2):577-584.), and even recommended in the physically active population (Kasper AM, Langan-Evans C, Hudson JF, Brownlee TE, Harper LD, Naughton RJ, Morton JP, Close GL. Come Back Skinfolds, All Is Forgiven: A Narrative Review of the Efficacy of Common Body Composition Methods in Applied Sports Practice. Nutrients. 2021, 13(4):1075.). This is because measurement by the currently popular BIA method requires adequate hydration of the body, which is often difficult to achieve in physically active people. We agree that the amount of adipose tissue assessed by measuring the thickness of the skinfolds relates primarily to subcutaneous fat. However, in young physically active women and men, adipose tissue accumulates mainly in this area of the body. Many years of observation of this population, which we have studied many times, confirm that the results obtained with the two methods are useful. In these already published articles, the assessment of body composition in young adults was performed using this method (Malara, Marzena, KÄ™ska, Anna, Tkaczyk, Joanna and LutosÅ‚awska, Grażyna. "Agreement of measures between measured body adiposity and calculated indices of fatness in sedentary and active male and female students" Biomedical Human Kinetics, 2022, 14(1), 271-279.; KÄ™ska A, Tkaczyk J, Malara M, IwaÅ„ska D. Metabolic Risk Factors in Young Men With Healthy Body Fat But Different Level of Physical Activity. Am J Mens Health. 2022, 16(1):15579883211070384.; Malara M, KÄ™ska A, Tkaczyk J, LutosÅ‚awska G. Metabolic Profile In Active Female Students Users And Non-Users Combined Oral Contraceptives. Ann Appl Sport Sci 2020; 8 (2).). But as the reviewer rightly pointed out, we use either method depending on the possibilities. In the present study, we used body composition data calculated on the basis of skinfolds thickness, as we wanted to obtain the largest possible groups (to carry out regression analysis). We analysed the results collected during three research projects that investigated the diet and body composition of female and male students. Skinfolds thickness measurements were taken in all projects.
Results:
• Authors can not compare effects of DII (DII not DII index as the name includes word ”index“) on body components as there is not enough data: only a single measurement of body composition was performed (not reliable measurement) and dietary data from only 1 week (even less: 4 days) were collected. Please remove this part of the paragraph.
- Authors haven’t present any results or parameters suggesting that diet influenced the inflammatory status of the participants (for instance: plasma levels of inflammatory markers before and after diet usage – authors mentioned about this fact in limitations of the study paragraph); please remove this statement;
Authors’ reply: We have discussed the above reviewer comments on the feasibility of regression analysis and how to interpret it with the statistician. We ascertained that with a single body composition measurement and a single dietary assessment and sufficiently large group sizes we could perform such an analysis. We can predict how quickly body composition (its components) will change in relation to changes in DII. We may not have written this very deftly in the Results, so we have made a correction to this paragraph.
Discussion:
• ‘’…diet of some students favored an increase in inflammatory markers” this sentence has no reflection in the results – remove or rephrase the sentence;
Authors’ reply: This sentence has been changed.
- It would be better not to make any assumptions concerning immunity, as, again, authors haven’t measured any parameters concerning immune system functioning.
Authors’ reply: It is true that in the present research, we did not assess the concentration of inflammatory markers, but studies confirmed the association between DII and such markers exist and were cited in manuscript. These studies show that we can assess the risk of inflammation based on DII.
- There are some results which are presented in results section but not discussed in discussion section: lack of differences in cholesterol consumption in females and saturated fatty acids in males – supplement the discussion;
Authors’ reply: Thank you for this comment. Discussion has been supplemented with commentary on these results.
Thank you again for your insightful review.

Reviewer 2 Report
Comments and Suggestions for Authors
A brief summary
The topic of the article is interesting with a potential scientific contribution. There are minor mistakes that should be checked and possibly changed.
General comments:
The article needs an English language check with a professional.
Please use the same font and size of the letters in the text. There are obvious errors in the text in some parts.
Specific comments:
Please, specify all abbreviations in the abstract.
After explaining the abbreviation for the first time, continue to use only abbreviations.
Last sentence in the Abstract is unnecessary.
Keywords are not listed correctly. Please use commas.
In the Introduction, explain in more detail the population on which the study was conducted.
Please specify the Ethical Approval Code in the Materials and Methods.
There are some typos in the text. Please correct them. f.e. in 2.4. - there are also unnecessary text parts. Authors should only describe their method here and refer to a specific reference in such a way that the method is modified. Please rewrite this paragraph.
It is enough to mention Table 1 in the text (in the Results section). There is no need to mention also in the end of the sentence.
Please use the same number of decimal places everywhere when specifying numbers.
Please change commas to dots in numbers on Figures 1 and 2.
First two sentences in the Discussion section are unnecessary.
In third paragraph in the Discussion section please find more appropriate expression instead “The start of studies”.
It is suggested to expand the Conclusion a bit.
Please check the References and correct typos.
Remove references older than 5 years unless they are specifically needed for the article.
Comments on the Quality of English Language
The article needs an English language check with a professional.
Author Response
Answer to Reviewer 2
We would like to thank the reviewer for his time and effort in reviewing our work and for all the feedback on the manuscript we submitted. We are grateful for the specific points the reviewer highlighted, which have enabled us to improve our manuscript.
The reviewer’s comments are outlined below with our point-by-point responses.
General comments:
The article needs an English language check with a professional.
Please use the same font and size of the letters in the text. There are obvious errors in the text in some parts.
Authors’ reply: The font was standardised and the manuscript was checked by a professional as suggested by the reviewer.
Specific comments:
Please, specify all abbreviations in the abstract.
Authors’ reply: All abbreviations have been specified.
After explaining the abbreviation for the first time, continue to use only abbreviations.
Authors’ reply: Applied throughout the manuscript.
Last sentence in the Abstract is unnecessary.
Authors’ reply: The mentioned sentence has been removed.
Keywords are not listed correctly. Please use commas.
Authors’ reply: This has been corrected.
In the Introduction, explain in more detail the population on which the study was conducted.
Authors’ reply: We have specified the description of the study population.
Please specify the Ethical Approval Code in the Materials and Methods.
Authors’ reply: The Ethical Approval Code has been added.
There are some typos in the text. Please correct them. f.e. in 2.4. - there are also unnecessary text parts. Authors should only describe their method here and refer to a specific reference in such a way that the method is modified. Please rewrite this paragraph.
Authors’ reply: In line with the reviewer's comment, we have removed unnecessary information from this paragraph.
It is enough to mention Table 1 in the text (in the Results section). There is no need to mention also in the end of the sentence.
Authors’ reply: The sentence regarding the content of Table 1 has been corrected as suggested by the reviewer.
Please use the same number of decimal places everywhere when specifying numbers.
Authors’ reply: Thank you for this comment. The presentation of the numerical values has been standardised.
Please change commas to dots in numbers on Figures 1 and 2.
Authors’ reply: It has been changed.
First two sentences in the Discussion section are unnecessary.
Authors’ reply: In line with the reviewer's comment, we have removed these sentences.
In third paragraph in the Discussion section please find more appropriate expression instead “The start of studies”.
Authors’ reply: This sentence has been corrected.
It is suggested to expand the Conclusion a bit.
Authors’ reply: The Conclusion has been completed.
Please check the References and correct typos.
Authors’ reply: Font has been standardised in this section.
Remove references older than 5 years unless they are specifically needed for the article.
Authors’ reply: Changes have been made to the references as recommended by the reviewer.

Round 2
Reviewer 1 Report
Comments and Suggestions for Authors
I still have some concerns about visceral fat content and measurement, but I can also accept the authors' replies. I have no other issues, so paper may be published in present form.